# Do Improvements in Maternal Mental Health Predict Improvements in Parenting? Mechanisms of the Mindful with Your Baby Training

**DOI:** 10.3390/ijerph19137571

**Published:** 2022-06-21

**Authors:** Eva S. Potharst, Manon Kuijl, Daphne Wind, Susan M. Bögels

**Affiliations:** 1UvA Minds, Academic Outpatient (Child and Adolescent) Treatment Centre of the University of Amsterdam, Banstraat 29, 1071 JW Amsterdam, The Netherlands; mkuijl@uvaminds.nl (M.K.); mindfulnessbydaphne@gmail.com (D.W.); 2Research Institute of Child Development and Education, University of Amsterdam, Nieuwe Achtergracht 127, 1018 WS Amsterdam, The Netherlands; 3Mindfulness by Daphne, Harlingerstraatweg 27, 8913 AB Leeuwarden, The Netherlands; 4Developmental Psychology, University of Amsterdam, Nieuwe Achtergracht 129-B, 1018 WS Amsterdam, The Netherlands; s.m.bogels@uva.nl

**Keywords:** mindfulness, mindful parenting, postpartum depression, mental health, parenting stress, parental self-efficacy, bonding, intervention, infants, mechanisms

## Abstract

Postpartum mental health symptoms are associated with parenting difficulties, which have negative consequences for child development. Interventions for young mothers should target their mental health problems and parenting difficulties. Mindful with Your Baby (MwyB) is an intervention for parents, with a baby, who experience (mental) health problems and/or stress or insecurity in parenting. This study seeks to replicate previous effects of MwyB regarding mindfulness, mindful parenting, maternal (mental) health (psychological distress, depressive mood, physical health complaints) and parenting outcomes (parenting stress, parental self-efficacy, bonding), and gain insight into the working mechanisms of the training. Mothers with babies aged 1–18 months (*n* = 61) completed questionnaires at waitlist, pretest, posttest, and 8-week follow-up. No significant differences were seen between the waitlist and pretest. Significant improvements in all outcomes were shown in the posttest (except for physical health complaints) and follow-up, compared to the pretest. Improvements in depressive symptoms and physical health complaints were dependent on improvements in mindfulness. Improvements in parental self-efficacy were dependent on improvements in mindful parenting. Improvements in some (mental) health and parenting outcomes seemed to be bidirectional. The results suggest that both mindfulness and mindful parenting are important for mothers who experience psychological distress and/or stress or insecurity in parenting their babies.

## 1. Introduction

Having a baby is a transformative process, bringing change in all aspects of life [1]. Becoming a mother can be one of the most rewarding and fulfilling experiences in adult life, but it requires a huge adaptation for which several personal, relational, and contextual resources are needed [2]. The availability of these resources on the one hand, and stress factors on the other hand, have a large impact on how new parenthood is experienced and whether well-being increases or decreases [2,3]. For a considerable number of parents, the postpartum period is coloured by feelings of stress, anxiety, and depression, and mothers, specifically, also suffer from physical health problems [1,4,5]. Although fathers also suffer more from mental health problems than in other periods of life, the problem of postpartum mental health problems seem to be highest in mothers [6,7].

Maternal mental health problems, defined as the presence of symptoms of stress, depression and/or anxiety, resulting in emotional suffering [8], can have negative consequences for child development, even if symptoms do not meet criteria for a disorder [8,9]. A review on parental postpartum mental illness and infant development highlighted the impact of postpartum mental health on several domains of early child development, such as regulatory behaviour, developmental milestones, and internalising and externalising behaviour [6]. A meta-analysis on maternal mental health and school-age child development showed that post-partum mental health problems increased the likelihood of suboptimal child cognitive, behavioural and socio-emotional development [10].

It has been hypothesised that the association between parental mental health problems and negative outcomes for children is mediated by difficulties in parenting [11]. Several studies have confirmed this hypothesis. In a community-based study, in which 1036 families participated, parental postnatal symptoms of depression were associated with the socio-emotional functioning of a child at 18 months, and this association was mediated by parenting stress [12]. Another study with a sample of 88 women who experienced childbirth-related distress showed that parenting stress mediated the association between the symptoms of post-traumatic stress at 3 months, and internalising and externalising behaviours at 18 months [13]. A study with a sample of 232 women living in a low-income community showed that a positive screening for depression at 2 months postpartum predicted infant at-risk social-emotional emotional development at 6 months, and that this association was mediated by low maternal bonding to the infant [14].

These studies not only shed light on the mediating role of difficulties in parenting, but also brought forward how difficulties in parenting can take many forms. One particular parenting difficulty is parenting stress. Parenting stress is defined as the imbalance between the demands that are associated with parenting and the perceived resources that are available to meet these demands [15]. Postpartum parenting stress is a risk factor for later socio-emotional problems [16]. A study on 31 mothers with a 2.5–5 year old child with a developmental disorder showed that the relationship between parental distress and child behaviour was mediated by the quality of the mother’s assistance (but not by two other measures of parenting, namely level of involvement and mother’s presence) [17]. Several studies have shown associations between postpartum mental health problems and parenting stress [18,19,20]. In a longitudinal study of 94 women, it was shown that both prenatal and postnatal depression and anxiety predicted parenting stress [19]. In another longitudinal study on risk factors for prenatal depression, postnatal depression, and parenting stress, the only significant risk factor for parenting stress was postnatal depression [20]. However, parenting stress also significantly predicted postnatal depression, suggesting that postnatal depression and parenting stress may be interrelated [20]. A complex relationship between depressive symptoms and parenting stress was found in a longitudinal study in which depressive symptoms and different aspects of parenting stress were measured at 3, 7, and 14 months postpartum in women who were selected as being at risk for postpartum depression [21]. Parenting stress was shown to predict later depressive symptoms, but depressive symptoms also predicted a specific aspect of parenting stress, namely difficult child stress [21]. Thus, it may be that parenting stress and mental health problems together form a vicious cycle that negatively impacts child development.

Another difficulty that mothers can face in the early period of parenthood is low parental self-efficacy [22]. Parental self-efficacy means parents’ faith in their ability to parent and influence their child in a health- and success-promoting manner [23]. Higher parental self-efficacy is predictive of better cognitive, behavioural, and socio-emotional development [23]. Postpartum mental health problems such as depressive and anxiety symptoms and parental self-efficacy have been shown to be negatively related [24,25,26]. A study on the association between history of childhood maltreatment and low parental self-efficacy in the postpartum period showed that this association was significant only for women with low well-being (depression or low resilience, which was defined in this study as tenacity, ability to tolerate negative affect, security of relationships, control, and spiritual influences) [27], suggesting that low well-being is a risk factor for low parental self-efficacy. However, other studies have shown that low parental self-efficacy in the postpartum period is a risk factor for depressive symptomatology [28,29]. In a review on the role of parental self-efficacy in parent and child adjustment, it was described that low parental self-efficacy can contribute to vulnerability to mental health problems (specifically depression), but that depression can also lead to lower parental self-efficacy, suggesting a transactional relationship [30].

A third difficulty in parenting that can play a large role in the early period of motherhood is poor maternal bonding with the infant [31]. Maternal bonding is defined as the emotional tie experienced by a mother towards her child [31]. Higher maternal bonding is predictive of higher infant attachment quality, easier infant temperament and more positive infant mood [32]. In terms of bonding, symptoms of depression, anxiety, stress, and post-traumatic stress are negatively related, and quality of life is positively related [33,34,35]. Maternal lifetime and current depression have been shown to be a risk factor for poor bonding [36], and maternal and paternal depressive symptoms at six weeks postpartum have been shown to be associated with poor bonding with the infant at six months postpartum [37]. The possibility that poor parental bonding might also lead to the development of depressive symptoms has hardly been studied [38,39].

Although effective treatment of psychological distress is important for any age group, it may be even more urgent for mothers with a baby, because of the serious consequences psychological distress may have in this period of life for the quality of parenting and infant development. However, focusing treatment on parental mental health problems alone may not be enough, as this is found not to be effective in improving parenting difficulties and the parent–child relationship [40,41]. In a systematic review on the effect of interventions for postpartum depression on the parent–child relationship and child outcomes, it was suggested that a decrease in depressive symptoms may be necessary but not sufficient to improve the parent–child relationship and child outcomes [42]. In a systematic review on the effect of perinatal depression treatment on parenting and child development, no definite conclusions could be drawn regarding a treatment with the potential to influence both maternal and infant outcomes [43]. Thus, treatment in the postpartum period should include interventions focused on mental health, parenting and the parent–child relationship.

An intervention with both intrapersonal and interpersonal potential for improvement is mindful parenting training [44,45]. Mindful parenting interventions extend mindfulness training, which focuses merely on intrapersonal experiences to the interpersonal context of parenting [46,47]. Mindfulness can be defined as “the awareness that emerges through paying attention, on purpose, in the present moment, and nonjudgmentally to the unfolding of experience moment by moment” [48], and mindful parenting as “paying attention to your child and your parenting in a particular way: intentionally, here and now, and non-judgmentally” [49]. Mindful parenting training has shown to decrease internalising and externalising psychopathology and physical health complaints in parents [50,51]. Moreover, it has been shown to improve parenting in several areas, such as parenting stress, parental over reactivity and experiential avoidance in parenting [50]. Aspects of observed parental behaviour and the parent–child relationship, namely sensitivity and acceptance, have also been shown to improve after mindful parenting training [52]. Child psychopathology also decreased after their parents followed mindful parenting training [52,53]. The favourable results regarding child outcomes may be associated with improvements in parental mindful parenting [51,53]. A study in which a mindful parenting training was offered in a child psychiatric setting showed that improvement in child psychopathology was dependent on improvement in mindful parenting (not general mindfulness), and improvement in parent psychopathology was dependent on improvement in general mindfulness (not mindful parenting) [53].

The possibility that an increase in mindful parenting abilities may have a positive effect on other areas of parenting is also illustrated by non-interventional studies. For example, a cross-sectional study of 333 parents with children in the age category of 8 to 18 years showed that higher mindful parenting mediated the relationship between higher mindfulness and lower parenting stress, authoritarian and permissive parenting styles, and higher levels of authoritative parenting [54]. In a large sample sized (564 parents with 3 to 17-year-old children) longitudinal study, it was shown that higher mindfulness was related to higher mindful parenting 4 months later, that higher mindful parenting was associated with higher levels of positive parenting and lower levels of negative parenting at 8 months, and that lower levels of negative parenting practices were related to lower child internalising and externalising symptoms at 12 months [55]. Another longitudinal study, in which a community sample of 246 parents of 3 to 12-year-old children participated, showed that higher levels of mindful parenting were positively related to positive emotion socialisation strategies and negatively related to negative emotion socialisation strategies [56]. An observational study (*n* = 37) in which mother-infant interactions were coded, showed that self-reported mindful parenting was associated with the mother’s gaze toward their child, which means that mothers scoring high on mindful parenting spent more time paying attention to their child [57]. In addition, specific aspects of mindful parenting were shown to be associated with other parenting and parent–child interaction outcomes, such as maternal sensitivity, maternal positive facial expression, and dyadic synchrony of positive effect [57].

To meet the needs of parents with infants, a specific mindful parenting programme was developed for this age group, named Mindful with Your Baby [58]. In this 8-week training programme, elements of general mindfulness interventions, such as mindfulness-based stress reduction (MBSR) [59], mindfulness-based cognitive therapy (MBCT) [60], and specific mindful parenting exercises [61], are taught. In order to support parents in integrating mindful parenting in contact with their baby, the sessions are given in the presence of the baby [58]. A study in a mental health care setting yielded positive results on areas such as maternal well-being, internalising and externalising psychopathology, parenting confidence, self-reported maternal responsivity, and infant positive affectivity [58]. In a non-clinical setting, a positive effect was shown regarding symptoms of depression, anxiety, general stress, and parental stress [62]. In an observational study, in which mother–child interaction observations were coded, positive effects were shown on maternal acceptance of the child, maternal mind-mindedness, and child responsivity [63].

To summarise, postpartum maternal mental health problems can have serious negative consequences for children’s development, and these negative consequences may be mediated by parenting difficulties such as parenting stress, low parental self-efficacy, and poor mother-infant bonding. It is important to treat mothers with babies who suffer from mental health problems, and to focus treatment not only on these problems, but also on parenting difficulties. Mindful with Your Baby is an intervention with potential for both intrapersonal (maternal [mental] health) and interpersonal (parenting) potential for improvement. The current study seeks to replicate previous studies that showed improvement in mindful parenting, mindfulness, maternal (mental) health, and parenting. Furthermore, the working mechanisms of the Mindful with Your Baby training are unknown. A meta-analysis on the working mechanisms of MBSR and MBCT showed that mindfulness is one of the mechanisms underlying the positive effects on mental health. Because mindfulness meditation and psychoeducation, similar to what is offered in MBSR and MBCT, are part of the Mindful with Your Baby training, also in Mindful with Your Baby, increased mindfulness may be the mechanism through which mental health improves. As non-intervention studies showed the predictive value of mindfulness on mindful parenting, also for the Mindful with Your Baby participants, improvement in mindfulness may (partly) underly improvements in mindful parenting. Mindful with Your Baby does not involve the teaching of specific parenting techniques. Therefore, positive changes in parenting could be attributed to improvement in mindfulness, mindful parenting, or mental health problems.

In this study, Mindful with Your Baby was offered to mothers who experienced psychological distress and/or stress or insecurity in parenting. The study seeks to replicate previous studies on the effects of the Mindful with Your Baby training regarding mindfulness and mindful parenting, (mental) health (psychological distress, depressive mood, physical health complaints), and parenting outcomes (parenting stress, parental self-efficacy, maternal bonding to the infant), and gain insight into the working mechanisms of the training. We hypothesised that the Mindful with Your Baby training would have a positive effect on mindfulness and mindful parenting, maternal (mental) health, and parenting outcomes. Furthermore, it was hypothesised that improvement in mindfulness between the pretest and posttest predicted improvement in mindful parenting in the follow-up. We expected that improvement in mindfulness (over and above improvement in mindful parenting) between the pretest and posttest predicted improvement in (mental) health outcomes in the follow-up. Improvement in mindful parenting (over and above improvement in mindfulness) between the pretest and posttest was expected to predict improvement in (other) parenting outcomes at follow-up. Lastly, improvement in (mental) health between the pretest and posttest was expected to predict improvement in parenting outcomes at follow-up. Hypotheses on the mediation of effects of Mindful with Your Baby have been summarised in Table 1, and have been visually represented in Figure 1.

## 2. Materials and Methods

### 2.1. Participants

Parents were referred to, or admitted themselves to, Mindful with Your Baby training. Seventy parents participated in the training, of which 65 parents participated in one of the 15 small groups (range 2–7 parents per group), four parents participated as a couple in an individual training (two individual trainings for two couples), and one parent participated individually without her partner. If parents decided to participate in the training, they were asked to also participate in the study. Of the 70 training participants, 63 (90.0%) decided to also participate in the study. Of those 63 parents, two (3.2%) were fathers. For the current study, the fathers were not included in the analyses. Thus, the final study group consisted of 61 mothers.

Sixty-one mothers (*M_age_* = 32.9 years; *SD* = 3.7) with their 0–18 month old babies (*M_age_* = 7.9 months; *SD* = 5.4), participated in the Mindful with Your Baby training and in the study because of psychological distress and/or stress or insecurity in parenting. Fifty mothers (82.0%) followed the training in an outpatient mental health care setting after referral from a general practitioner or psychologist, and eleven (18.0%) mothers admitted themselves to the Mindful with Your Baby training in a preventive setting. In case mothers were referred for the training, the costs were covered by the municipality or insurance. When mothers followed the training in a preventive setting, depending on the family income, they received a discount on, or refund of, the training costs if they participated in the research project, which was funded by the MindMore Foundation. Sociodemographic characteristics of the participants are shown in Table 2.

### 2.2. Procedure

The data for this study was derived from an ongoing study, which focuses on the comparison of Mindful with Your Baby in a clinical and preventive setting. The current study was approved by the Ethics Committee of the University of Amsterdam (2017-CDE-7946). Written informed consent was obtained from all participants. All measurement occasions consisted of questionnaires that could be filled in online. Only mothers who were referred or admitted to the training at least one month prior to the start of the training (and who thus had to wait at least one month for the training to start), were invited to complete a waitlist assessment, to control for factors such as time, participation in research, and hope for change. Thirteen (21.3%) participants completed waitlist assessment (average waiting time between the completion of the waitlist assessment and the start of the training was 45.9 days, *SD* = 13.5, range 19–67 days). The pretest was administered one week prior to the start of the training and was completed by 58 (95.1%) of the mothers. The posttest was administered immediately after the training and was completed by 50 (82.0%) of the mothers. Lastly, the mothers received the follow-up assessment eight weeks after the training, and this assessment occasion was completed by 36 (59.0%) of the mothers. Some of the mothers also participated in additional research activities, namely parent–child observations and/or daily measurements, but these additional measures were not included in the current study. A flow diagram of participation in the study can be found in Figure 2.

### 2.3. Intervention

The Mindful with Your Baby training was conducted from May 2017 until March 2022. The training consists of 8 weekly 2-h sessions and a follow-up session 8 weeks after the training. The babies are present during the training, except for the first and fifth sessions, in which enough space is needed for the introduction and for the theme of self-compassion, respectively. The other sessions do have a programme with several exercises, but at the same time, flexibility is built in so that if stress arises in (one of) the mother–baby dyads, a short mindfulness meditation can be practiced with the group, namely a parent–child breathing space. This is an adjustment of the 3-min breathing space [60], in which parents not only practice with allowing their own feelings, but also with allowing the feelings of their baby. The training sessions are given by both a trainer and an assistant trainer, which is especially important during the formal meditations because the assistant trainer can keep her eyes open to be available for the babies and make sure they are safe, both physically and mentally.

The build-up and exercises of the training are based on the Mindful Parenting training [61]. Furthermore, the training is informed by the Infant Mental Health framework [64], which means that the developing relationship between mother and child is of primary importance in the training, and that the role that stress plays in the relationship between mother and child is recognized. During the training, mothers are taught mindfulness to regulate stress within the parent–child relationship. By learning to do short meditations in the presence of their baby, they learn that, as a mother, they can stay in touch with their own experience and take care of themselves. By doing watching meditations with their attention on their baby, they learn to stay present with their baby with a beginner’s mind, open to the signals their baby gives. Each session has its own theme: (1) becoming aware of the automatic pilot, (2) mindfully observing your baby, (3) Creating space for yourself, (4) responding sensitively to your baby, (5) taking care of yourself in difficult moments, (6) closeness and distance, (7) dealing with expectations, and (8) mindful parenting, each time, beginning anew. The practices and psychoeducation correspond to the themes of the session. Each sessions starts with a 10-to-15-min formal meditation practice, such as a sitting meditation with attention for the breath and body, followed by inquiry. This inquiry is focused on experiences during the meditation, among which is the way in which the parent divided her attention between herself and her baby. After the inquiry, experiences with the homework are discussed. In some sessions, there is another short practice before the break, such as the 3-min breathing space meditation. In the break, there is space for the mothers to talk with each other, drink something, and for example, to feed their baby. After the break, an exercise is done that is focused on the theme of the session, such as visualisation on a moment that the mother felt a distance in the contact between herself and her baby, and on how the mother can reconnect with her baby again after such a moment. The last exercise of the sessions with the babies present is a watching meditation, in which mothers practice observing their baby with full attention and with curiosity, to also notice what they experience while doing this, to reflect on what the baby might be experiencing, and some meditation instructions are also given around the theme of the session. In between the sessions, mothers do both formal and informal practices, mindful parenting exercises, and they read two readings on mindfulness or infant mental health subjects in the Mindful with Your Baby workbook [65], such as “The seven attitudes” [59], “Supporting your baby’s autonomy” [49], and the “Circle of Security” [66]. In the follow-up session, the mothers are reminded of all the themes of the previous sessions. A table which summarises the main and secondary themes, the session practices, the home practices and readings of each session has been published in a previous study on Mindful with Your Baby [62].

Before the COVID-19 pandemic, all training was given on location, which was an outpatient mental health care centre for the majority of the participants, or, for the mother-baby dyads who followed the training in a preventive setting, a mindfulness training centre, or a space rented by a Mindful with Your Baby teacher. During the pandemic, training was given online. Twenty-two parents (36.1%) followed the training online, and thirty-nine (63.9%) on location. All training was given by Mindful with Your Baby trainers (EP, DW or one of the trainers mentioned in the acknowledgements), who were also mindfulness trainers or mindful parenting trainers, and they were assisted by a psychologist, or a psychology or pedagogy intern.

### 2.4. Measures

*Mindfulness* was measured with the short form of the Dutch version of the Five Facet Mindfulness Questionnaire (FFMQ) [67,68]. The self-report questionnaire measures five facets of mindfulness: observing, describing, acting with awareness, non-judging, and non-reactivity. The short form consists of 24 items out of the original 39 items shown in a five-factor structure, similar to the original form. In the current study, only the full scale was used. An example of an item is: “I tell myself I should not be feeling the way I am feeling”. The items were scored on a 5-point Likert scale, ranging from 1 (never or very rarely true) to 5 (very often or always true). Higher scores indicate a greater mindfulness. The psychometric properties of the original scale were good in both a meditating sample and a non-meditating sample [68]. Cronbach’s alphas of the total scale in the current study were 0.90 at waitlist, 0.88 at pretest, 0.91 at posttest, and 0.93 at follow-up.

*Mindful parenting* was assessed with the Dutch version of the Interpersonal Mindfulness in Parenting Scale (IM-P-NL) [69,70]. The Dutch version of the IM-P contains 29-items that are scored on a 5-point Likert scale, ranging from 1 (never true) to 5 (always true). Higher scores reflect a higher level of mindfulness in parenting. Some small adaptions were made in the formulation of some items to make them applicable for mothers with a baby. For example, “child” was changed into “baby” and “parenting” or “raising” to “nurturing”. Three items (items 4, 8 and 28) were inapplicable for mothers with a baby, and therefore were left out for the current study. An example of an item was, “I am often so busy thinking about other things that I realize I am not really paying attention to my baby”. The original IM-P comprises five hypothesised subscales that were not factor analytically validated. A Dutch validation study conducted a factor analysis that revealed satisfactory reliability for these six subscales [70]: listening with full attention, compassion for the child, non-judgmental acceptance of parental functioning, emotional non-reactivity in parenting, emotional awareness of the child, and emotional awareness of self. In the current study, only the full scale was used. The Cronbach’s alphas of the total scale in the current study were 0.88, 0.91, 0.91, and 0.90 at waitlist, pretest, posttest, and follow-up, respectively.

*Psychological distress* was assessed with the Dutch version of the outcome questionnaire (OQ) [71,72]. The OQ is a self-report questionnaire and comprises 45 Likert scale items on a 5-point scale, ranging from 0 (never) to 4 (almost always) that are divided into four subscales. The Dutch version of the OQ consists of four analytically validated subscales [71]. The subscale Symptom Distress measures the symptoms of depression, anxiety, and substance dependence. The subscale Anxiety and Symptomatic Distress measures problems of anxiety and the accompanying physical characteristics of anxiety. The subscale Interpersonal Relations measures problems encountered in interpersonal relations. The subscale Social Role measures distress on a broader social level, including distress encountered at work, during education, and during leisure activities. In the current study, only the full scale is used. Higher scores on a scale indicate more distress symptoms. A Dutch study revealed that the Dutch version of the OQ showed satisfactory reliability and validity [72]. The Cronbach’s alphas of the total scale in the current study were 0.94, 0.93, 0.93, and 0.94 at waitlist, pretest, posttest and follow-up, respectively.

*Depressive mood*, *physical health complaints*, *and parenting stress* was assessed with the Parenting Stress Questionnaire (PSQ, in Dutch: Opvoedingsbelastingvragenlijst [OBVL]) [73]. The self-report questionnaire contains 34 questions that are scored on a 4-point Likert scale, ranging from 1 (not true) to 4 (very true) and can be divided into five subscales: depressive mood, physical health complaints, parent–child relationship problems, parenting problems, and parental role restriction. In the current study, the subscales depressive mood and physical health complaints are reported separately, while the three parenting-related subscales were combined into one parenting stress scale. Higher scores on a scale indicates more problems in the (sub)scale area. The OBVL showed a satisfactory reliability and validity [73]. Cronbach’s alphas at pretest, posttest and follow-up in the current study were 0.83, 0.82, 0.87, and 0.88 for the subscale depressive mood at waitlist, pretest, posttest and follow-up, respectively, 0.88, 0.83, 0.87, and 0.80 subscale physical health complaints at waitlist, pretest, posttest, and follow-up, respectively, and 0.93, 0.89, 0.90, and 0.88 for the combined parenting stress scale at waitlist, pretest, posttest, and follow-up, respectively.

*Self-efficacy in the nurturing role* was measured with the Dutch version of the Self-Efficacy in the Nurturing Role scale, postnatal version (SENR) [74,75]. The self-report questionnaire consists of 16 Likert-scale items on a 7-point scale, ranging from 1 (not at all representative of me) to 7 (strongly representative of me). The total score gives an indication of the mothers’ perceptions of their competence on basic skills required in caring for an infant, where a higher score indicates greater feeling of efficacy. The reliability of the Dutch version has been found to be high [76]. The Cronbach’s alpha in the current study were 0.80, 0.83, 0.86, and 0.85 at waitlist, pretest, posttest, and follow-up, respectively.

*Mother-infant bonding* was measured with the Pre and Postnatal Bonding Scale (PPBS) [77]. The PPBS is a self-report scale and comprises five items of positive feelings that a mother might feel toward her child. Mothers can rate the items on a 7-point Likert scale, ranging from 0 (not at all) to 3 (very much). Higher scores indicate more positive feelings of bonding. The PPBS showed adequate psychometric properties [77]. The Cronbach’s alphas in the current study were 0.85, 0.87, 0.92, and 0.85 at waitlist, pretest, posttest, and follow-up, respectively.

### 2.5. Data Analysis

All analyses were done with SPSS 24 software (IBM, Armonk, NY, USA). Differences between drop-outs and participants that finished the training were tested with *t*-tests and chi-square tests. Inspection of distribution of scores at all three measurement points indicated sufficient normality (tested with Shapiro–Wilk test), skewness and kurtosis <|3| of all variables [78], except for IM-P total score (*SW* = 0.947, *df* = 58, *p* = 0.013) and the PBBS (*SW* = 0.907, *df* = 60, *p* < 0.000) at pretest. The distribution of the PBBS and OBVL subscale health complaints showed insufficient normality at posttest (*SW* = 0.812, *df* = 49, *p* < 0.000 and posttest: *SW* = 0.877, *df* = 48, *p* < 0.000, respectively) and follow-up (*SW* = 0. 775, *df* = 34, *p* < 0.000 and *SW* = 0.912, *df* = 34, *p* = 0.010, respectively). Homogeneity of the variances was not violated for any variable, nor was the assumption of linearity. Finally, the variance of the residuals was equal (homoscedasticity).

The effects of the training were tested with multilevel regression models that are known to accommodate missing data [79]. The structure of the multilevel models for all outcomes consisted of the repeated measurements of these outcomes across the three measurement points nested within the participants. Measurement points were dummy coded with the pretest scores as a reference. The intercept was a fixed effect in all models.

To test hypotheses on the predictive value of improvement on one measure between the pretest and posttest (e.g., mindfulness) on improvement on another measure at follow-up (e.g., depressive symptoms), a difference score of the first measure (mindfulness at posttest minus mindfulness at pretest) was calculated, and added as an independent variable to the multilevel model testing change in the second measure (depressive symptoms). Moreover, interaction terms (difference score * posttest and difference score * follow-up) were added to the model. Only if the second interaction term (difference score * follow) was significant, was it regarded as significant predictive value of change in the first measure between the pretest and postest on change in the other measure at follow-up. To check for possible bidirectional effects, we repeated all analyses “vice versa” by including the independent variable as a dependent variable and the dependent variable as the independent variable. In the analyses in which improvement in mindfulness was the independent variable, and in which a significant effect was found, it was checked if this effect was still significant after adding improvement in mindful parenting. In addition, in the analyses in which improvement in mindful parenting was the independent variable, and when a significant effect was found, it was checked if this effect was still significant after adding improvement in mindfulness.

An effect was found to be significant when *p* < 0.05. All scores on the outcomes and difference scores were standardised across assessments, so that estimates of regression coefficients for dichotomous explanatory variables (measurement point) can be interpreted similarly to Cohen’s *d* effect sizes (0.2 small, 0.5 medium, 0.8 large; Cohen) [80], and estimates of regression coefficients for continuous explanatory variables (interaction terms of difference scores * measurement point) can be interpreted similarly to Pearson *r* effect sizes (0.1 small, 0.3 medium, 0.5 large; Cohen) [80].

## 3. Results

### 3.1. Drop-Out Analysis

Four mothers (6.6%) dropped out of the training. T-tests indicated no significant difference in the outcome measures at pretest between participants who completed the training and those who dropped out. Regarding sociodemographic characteristics, no differences were found between participants who completed the training and those who did not, with the exception of the sex of the baby, where the sex of the baby of the drop-outs was female in all cases.

Research drop-out was defined as not completing both the posttest and follow-up. Ten (16%) of the mothers were research drop-outs. There were no differences in outcomes at pretest between research drop-outs and mothers that were still participating in the research after pretest. There was one difference regarding sociodemographic characteristics: mothers who were research drop-outs were more often on sick leave (5; 50.0% versus 11; 21.6% in the mothers who were still participating in the research after pretest).

### 3.2. Effects of the Mindful with Your Baby Training

The means and standard deviations of all outcome measures at all measurement points are displayed in Table 3. The results of multilevel models of treatment outcomes predicted by measurement point are displayed in Table 4. No significant improvements were found between the waitlist and pretest. Significant improvements in mindfulness and mindful parenting were found at posttest (moderate and large effect sizes, respectively) and follow-up (large effect sizes). Furthermore, a significant decrease at posttest in psychological distress (small effect size), and depressive mood (moderate effect size) was shown, but no significant effect was shown at posttest for physical health complaints. However, all three (mental) health outcomes were shown to have significantly decreased at follow-up (moderate effect sizes). Parental stress significantly decreased (moderate effect size), parental self-efficacy and parental bonding significantly increased (small effect sizes) at posttest, and all three parenting outcomes showed an improvement at follow-up (moderate effect sizes).

### 3.3. Does Mindfulness Predict Mindful Parenting?

As was expected, improvement in mindfulness between the pretest and posttest was a significant predictor of improvement in mindful parenting at follow-up (*β* (*SE*) = 0.28 (0.13), *t* (40) = 2.20, *p* = 0.034, 95% CI [0.02, 0.57], small to moderate effect size).

To check for bidirectionality, it was tested whether improvement in mindful parenting between the pretest and posttest was a significant predictor for improvement in mindfulness at follow-up. This indeed was shown to be the case (*β* (*SE*) = 0.40 (0.16), *t* (36) = 2.56, *p* = 0.016, 95% CI [0.08, 0.72], moderate effect size).

### 3.4. Does Mindfulness Predict Maternal (Mental) Health?

As was expected, improvement in mindfulness between the pretest and posttest was significantly predictive of improvement in depressive mood (*β* (*SE*) = −0.65 (0.15), *t* (38) = −4.39, *p* < 0.001, 95% CI [−0.95, −0.35], large effect size) and physical health complaints (*β* (*SE*) = −0.46 (0.12), *t* (46) = −3.77, *p* < 0.001, 95% CI [−0.70, −0.21], moderate effect size) at follow-up. To check whether the effects were specific for mindfulness or mindful parenting would also be a significant predictor, improvement in mindful parenting was added to the model. Mindfulness was still the only significant predictor of improvement in depressive mood and physical health complaints at follow-up. Improvement in mindfulness was not a significant predictor of psychological distress (*p* = 0.196).

To check for bidirectionality, we tested whether improvement in (mental) health between the pretest and posttest was predictive of improvement in mindfulness at follow-up. Improvement in psychological distress, depressive mood, and physical health complaints between the pretest and posttest were not predictive of improvement in mindfulness at follow-up (*p* = 0.855, *p* = 0.272, *p* = 0.562, respectively).

### 3.5. Does Mindful Parenting Predict Other Parenting Outcomes?

The results of the multilevel analyses showed that improvement in mindful parenting between the pretest and posttest was a significant predictor of improvement in parental self-efficacy at follow-up (*β* (*SE*) = 0.52 (0.16), *t* (28) = 3.36, *p* = 0.002, 95% CI [0.20, 0.84], large effect size). To check whether the effects were specific to mindful parenting or whether mindfulness would also be a significant predictor, improvement in mindfulness was added to the model. When improvement in mindfulness was added to the model, improvement in mindful parenting was still the only significant predictor of improvement in parental self-efficacy. Improvement in mindful parenting between pretest and posttest was not found to be a significant predictor of parenting stress (*p* = 0.057) or maternal bonding (*p* = 0.249) at follow-up.

To check for bidirectionality, it was tested whether improvement in the three other parenting outcomes between pretest and posttest were significant predictors of improvement in mindful parenting at follow-up. Improvement in maternal bonding was found to be a significant predictor of improvement in mindful parenting (*β* (*SE*) = 0.34 (0.13), *t* (41) = 2.62, *p* = 0.012, 95% CI [0.61, 0.08], moderate effect size). However, this was not the case for parenting stress (*p* = 0.186) and parental self-efficacy (*p* = 0.212).

### 3.6. Does Maternal (Mental) Health Predict Parenting Outcomes?

In contrast to expectations, multilevel analyses showed that a decrease in psychological distress between the pretest and posttest did not significantly improve other parenting outcomes, including maternal parenting stress (*p* = 0.170), parental self-efficacy (*p* = 0.152), and bonding (*p* = 0.105) at follow-up. Improvement in maternal depressive mood between pretest and posttest was found to be a significant predictor of improvement in parental self-efficacy at follow-up (*β* (*SE*) = −0.45 (0.19), *t* (37) = −2.34, *p* = 0.025, 95% CI [−0.83, −0.06], moderate effect size). Improvement in depressive mood was not found to be a significant predictor for improvement in parenting stress (*p* = 0.096), and maternal bonding (*p* = 0.227) at follow-up. Improvement in physical health complaints between pretest and posttest also significantly predicted improvement in parental self-efficacy at follow-up (*β* (*SE*) = −0.52 (0.23), *t* (34) = −2.32, *p* = 0.027, 95% CI [0.98, −0.06], large effect size), but not in parenting stress (*p* = 0.337), and maternal bonding (*p* = 0.772).

To check for bidirectionality, it was tested whether improvement in the three parenting outcomes between pretest and posttest were significant predictors of improvement in (mental) health at follow-up. Improvement in parental self-efficacy between the pretest and posttest significantly predicted improvement in maternal depressive mood (*β* (*SE*) = −0.60 (0.16), *t* (27) = −3.73, *p* = 0.001, 95% CI [−0.93, −0.27], large effect size) and physical health complaints (*β* (*SE*) = −0.46 (0.14), *t* (28) = −3.39, *p* = 0.002, 95% CI [−0.73, −0.18], moderate effect size) at follow-up. Improvement in parental self-efficacy did not significantly predict improvement in psychological distress (*p* = 0.154). It was also examined whether improvement in parenting stress between the pretest and posttest predicted improvement in psychological distress, depressive mood, and health complaints at follow-up. Indeed, improvement in parenting stress between the pretest and posttest predicted an improvement in physical health complaints at follow-up (*β* (*SE*) = −0.29 (0.13), *t* (47) = 2.17, *p* = 0.035, 95% CI [0.02, 0.56], small to moderate effect size). However, this was not the case for psychological distress and depressive mood (*p* = 0.054, *p* = 0.243, respectively). Finally, the analyses showed that improvement in maternal bonding between the pretest and posttest was a significant predictor for improvement in depressive mood (*β* (*SE*) = −0.50 (0.17), *t* (40) = −3.04, *p* = 0.004, 95% CI [−0.17, −0.84], large effect size) and physical health complaints (*β* (*SE*) = −0.34 (0.13), *t* (48) = 2.72, *p* = 0.009, 95% CI [−0.09, −0.63], moderate effect size) at follow-up, but not for psychological distress (*p* = 0.233).

Figure 3 displays a summary of the significant results that were described in Section 3.3, Section 3.4, Section 3.5 and Section 3.6.

## 4. Discussion

The current study sought, firstly, to replicate previous studies on the effects of the Mindful with Your Baby training regarding mindfulness and mindful parenting, maternal (mental) health (psychological distress, depressive mood, physical health complaints), and parenting outcomes (parenting stress, parental self-efficacy, maternal bonding to the infant), and secondly, gain insight into the working mechanisms of the training. With respect to the effects of Mindful with Your Baby it was found that, compared to the pretest, no significant improvements were seen between the waitlist and pretest, while significant and substantial improvements on all outcome measures was shown at both the posttest and follow-up, except for physical health complaints, which only decreased significantly at follow-up. With respect to the working mechanisms, improvement in mindfulness between the pretest and posttest predicted improvement in mindful parenting at follow-up, but the same was shown for the inverse relationship: improvement in mindful parenting also predicted improvement in mindfulness. Improvement in mindfulness between the pretest and posttest predicted aspects of (mental) health at follow-up, namely depressive mood and physical health complaints, but not psychological distress. Improvement in mindful parenting between the pretest and posttest predicted one aspect of parenting, namely parenting self-efficacy, but not parenting stress and mother-to-infant bonding. Improvement in bonding between the pretest and posttest did predict improvement in mindful parenting at follow-up. Improvement in aspects of (mental) health between the pretest and posttest predicted improvement in parental self-efficacy, but not in parenting stress or bonding. Improvement on all three parenting outcomes between the pretest and posttest predicted improvement in (mental) health at follow-up.

The Mindful with Your Baby training is a mindfulness-based intervention that includes mindfulness meditations and psychoeducation. The training is offered in a setting that involves the presence of the babies for a large part, who may distract the parents during the training. The results of this study show that Mindful with Your Baby seems to be effective in improving mindfulness. This underscores the assumption that mindfulness does not need to be taught in a distraction–free environment, but that it can be practised in a challenging situation.

Positive results were also shown regarding mindful parenting and other aspects of parenting as well as aspects of maternal (mental) health. These findings replicate earlier studies in which a combination of improvement in (mental) health outcomes and parenting outcomes were shown [58,62]. Several reviews on interventions for mothers, with a baby, who experience mental health problems (specifically depression) underscore the importance of focusing the treatment on both the depressive symptoms itself and the difficulties in parenting and the parent–child relationship, and conclude that most interventions are not effective in treating both [40,41,42,43].

As expected, a greater improvement in mindfulness between pretest and posttest predicted greater improvement in mindful parenting at follow-up (small to moderate effect size). This is in line with theory on mindfulness and mindful parenting, that describes mindfulness as a basis for the development of mindful parenting [46,49]. The results are also in line with the results of a non-intervention study that showed that mindfulness in parents predicted later mindful parenting [55]. However, in the current study the inverse relationship was also found, even with a somewhat larger effect size: a greater improvement in mindful parenting predicted a greater improvement in mindfulness at follow-up (moderate effect size). Possibly, this result can be explained by the Motherhood Constellation [81], which is a new mother’s mental organisation, that mainly encompasses the well-being of her baby, and her own ability to care for and bond with her baby. When something is not right with the baby, and when the mother doubts her mothering abilities or her ability to form a bond with her baby, this gives rise to stress. Only when she becomes more secure about these issues, will she find space to attend more to herself and her own needs. Possibly, when a mother notices that she is more able to be there for her baby by becoming more mindful in her parenting, this creates more space to also attend to herself again, and supports her in her process of becoming more mindful in general.

As expected, we found that a greater improvement in mindfulness between the pretest and posttest predicted a greater decrease in certain (mental) health problems, namely depressive mood (large effect size) and physical health problems (moderate effect size). This is in line with a systematic review that found mindfulness to be a mechanism of action that leads to beneficial physical and psychological outcomes in people with physical and/or psychological conditions [82]. However, in the current study, we did not see a significant association between improvement in mindfulness between the pretest and posttest and psychological distress at follow-up. Possibly, this was due to the broadness of the psychological distress measure that we used: besides measuring general symptom distress and anxiety and symptomatic distress, the OQ also measures interpersonal relations and social role. An increase in mindfulness may not have such a large effect on these aspects, also given the specific period the participants were in and what their attention was focused on in this period (learning to take care of their babies; to understand their baby; to parent their baby; integrating the baby in their partner relationship, family, working and social life; and taking care of themselves in the midst of this all).

We found that a greater improvement in mindful parenting between the pretest and posttest significantly predict a greater increase in parental self-efficacy (large effect size), while it did not significantly predict greater decrease in parenting stress (but did approach significance, moderate effect size). This difference in effect is surprising, as much more research has been done on the effect of mindful parenting (training) on parenting stress e.g., [83], than on parental self-efficacy [84]. Improvement in mindful parenting between the pretest and posttest did not predict improvement in mother-to-infant bonding. This contradicts the findings of a study that was carried out during the COVID-19 pandemic in mothers with 0 to 12 month old babies, that showed that higher mindful parenting at T1 predicted less impaired mother-to-infant bonding at T2 [85]. In the current study, we found a reverse relationship: a greater improvement in bonding between pretest and posttest predicted greater improvement in mindful parenting at follow-up. This finding suggests that the more a mother has positive feelings in relation to her baby, the easier it is for her to stay present with her baby, become aware of the baby’s experience, feel compassionate with her baby, and be non-reactive in parenting.

Our expectation that improvement in (mental) health between the pretest and posttest would predict improvement in parenting at posttest was only partially confirmed. A decrease in psychological distress between the pretest and posttest was not predictive of improvements in parenting, but a greater decrease in depressive mood and physical health complaints was predictive of a greater improvement in parental self-efficacy (moderate and large effect sizes, respectively). Our expectation to find transactional relationships was also partially confirmed as greater improvements in parental self-efficacy predicted a further decrease in depressive mood and physical health complaints (large and moderate effect sizes, respectively). This result implies that as a mother starts to feel more confident in taking care of her baby, her depressive and physical health complaints wane, and this helps her in feeling she can take good care of her baby. Such a transactional relationship was also suggested in a review on parental self-efficacy, which showed that depression could lead to low parental self-efficacy, but also that low parental self-efficacy could make a parent more vulnerable to depressive symptoms [30]. In a study on postpartum depression in mothers and fathers, it was found that parents with lower parental self-efficacy had greater risk of developing a postpartum depression [86]. The current study adds that not only is low parental self-efficacy a risk factor for developing postpartum depression, but also that helping mothers grow in their parental self-efficacy may alleviate depressive symptoms.

Greater improvements in depressive mood and physical health complaints between the pretest and posttest were not associated with a greater decrease in parenting stress or a greater increase in feelings of bonding. This is not in line with earlier research, which showed maternal mental health to be a significant predictor of parenting stress and bonding [19,35]. However, we did find reverse relationships. First, a greater decrease in parenting stress between pretest and posttest predicted a greater decrease in physical health complaints at follow-up (small to moderate effect size). The idea that a decrease in stress could be beneficial for patients with chronic pain has given rise to the development of MBSR [87]. The current study suggests that also diminishing a specific form of stress, namely parenting stress, possibly helps in alleviating physical health problems in mothers with babies, or recovering from complaints that were still related to pregnancy, giving birth, or breastfeeding. Second, a greater improvement in mother-to-infant bonding between the pretest and posttest predicted a greater decrease in depressive mood and physical health complaints at follow-up (large and moderate effect size, respectively). This finding is surprising, given the large number of scientific studies on the effect of depressive symptomatology on bonding, but the scarce evidence that there might also be a reverse relationship [38,39]. This result underscores the importance of attending to the mother–child relationship when interventions are offered to women with postpartum mental health complaints. This may not only be important to improve parenting and the parent–child relationship, but also may reduce the depressive symptoms of the mother. For mothers in the early stages of mothering, how they feel about themselves and their future may largely depend on how effective they feel in their role of being a mother to their young child.

All in all, we found more evidence for the beneficial effect of improving parenting on maternal (mental) health, than for the beneficial effect of improving maternal (mental) health on parenting. Two reviews on treatment for postpartum depression concluded that focusing on the depression alone is not sufficient to improve parenting and the parent–child relationship [40,41]. The results of the current study suggest that it may be that focusing on the treatment of the depression alone may not only have no benefits for parenting and the parent–child relationship, but that the depressive symptoms will also be treated more effectively if attention is given to parenting and the parent–child relationship. This conclusion, however, should be seen in the context of a mindful parenting intervention, in which mothers are invited to explore their relationship with their baby from a perspective of mindfulness.

The predictive value of mother-to-infant bonding for maternal (mental) health and other aspects of parenting might be an interesting focus for future research. Previous research on bonding assessed maternal mental health factors primarily as antecedents of bonding [31]. When the consequences of bonding were studied, research focused on infant development [32]. Because the current research suggests that bonding may also have an effect on maternal depressive mood, physical health complaints, and on mindful parenting, it may be worth broadening the research on the consequences of bonding. Future research could also compare the effects of general mindfulness training for parents (e.g., MBSR [59] or MBCT [60]), and parenting-specific mindful training for parents (e.g., mindful parenting training [61] or Mindful with Your Baby). Because the current study shows differential effects of mindfulness and mindful parenting on several outcomes, it may be important to explicitly include mindful parenting exercises in a mindfulness training programme for parents. Furthermore, a similar study as the current study could be done with observational measures of parenting instead of self-reported parenting measures. Future research should also address the differences in effect between the face-to-face and online versions of Mindful with Your Baby. Future studies on the working mechanisms of the Mindful with Your Baby training could also include a measure of anger and withdrawal. A qualitative study on postpartum mental health and parenting showed that the presence of mental health symptomatology in general and specifically parental anger were associated with more negative accounts of the self, the baby, and parenting [88]. Earlier research on the effectiveness of Mindful with Your Baby showed a decrease in externalising psychopathology after the training, but it is unclear if anger or externalising psychopathology plays a role in improving the parenting outcomes. In a review on the pathways that are associated with maternal withdrawal, it was concluded that maternal withdrawal was a risk factor for later psychopathology in the children [89]. Earlier research on the effects of Mindful with Your Baby training on parent–child interaction did not include a measure of withdrawal [89].

The current study had several strengths and limitations. A strength was that it did not just focus on effectiveness but also looked at mechanisms of change. Another strength was that we looked at different aspects of maternal (mental) health and parenting. A limitation was the reliance on self-report measures only, especially regarding the measurement of aspects of parenting. Another limitation was the focus on mothers only; just two fathers had participated in the Mindful with You Baby training, which was too little to be able to include them and assess possible differences between mothers and fathers. A third limitation was the quasi-random waitlist. A fourth limitation was the power for the analyses on mechanisms of change. Some associations that were of moderate effect size did not reach significance, which may point to limited power.

## 5. Conclusions

The results are in line with earlier studies showing Mindful with Your Baby seems to be effective in improving mindfulness, mindful parenting, maternal (mental) health, and parenting. Improvements in mindfulness had a positive effect on mindful parenting and aspects of maternal (mental) health, while improvements in mindful parenting had a positive effect on improvements in mindfulness, and parenting outcomes. Improvements in maternal (mental) health had a positive effect on just one parenting outcome, namely, parental self-efficacy, while improvements in all three parenting outcomes had positive effects on maternal (mental) health outcomes.

The results of the current study suggest that both mindfulness and mindful parenting skills are important to learn for mothers with (mental) health problems and/or stress or insecurity in parenting. Improved mindfulness may be beneficial for maternal (mental) health, and improved mindful parenting may be beneficial for the quality of parenting. Furthermore, this study underscores the importance of attending to parenting in the treatment of postpartum mental health problems.

## Figures and Tables

**Figure 1 ijerph-19-07571-f001:**
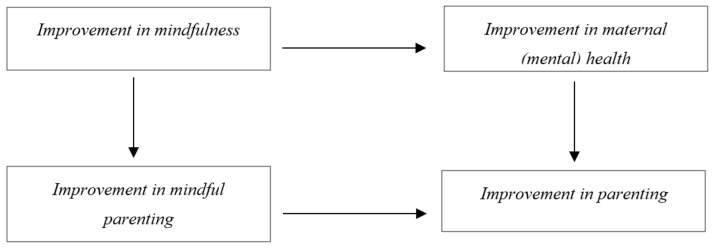
Hypotheses on the working mechanisms of Mindful with Your Baby.

**Figure 2 ijerph-19-07571-f002:**
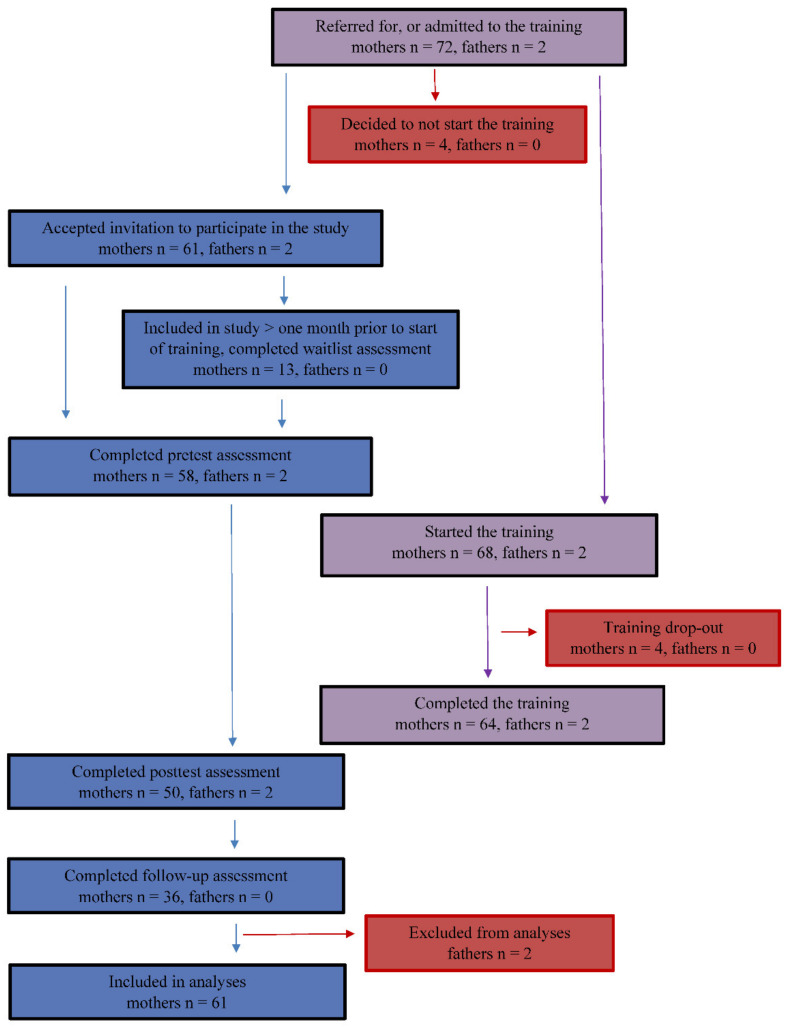
Flow diagram.

**Figure 3 ijerph-19-07571-f003:**
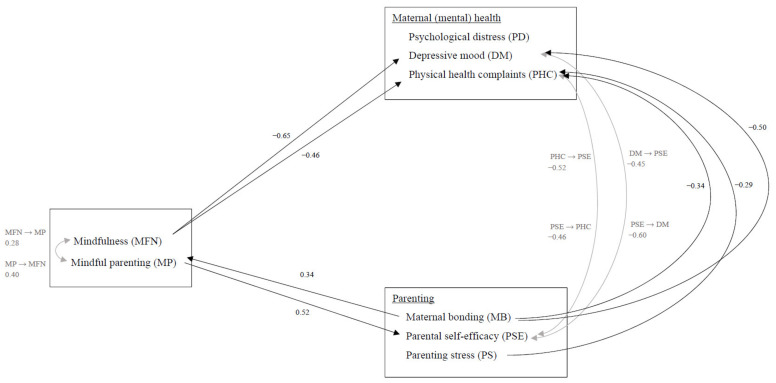
Visual representation of the predictors of treatment outcomes with effect sizes (grey lines for bidirectional effects).

**Table 1 ijerph-19-07571-t001:** Hypotheses on mediation of effects of Mindful with Your Baby, including the timepoints.

Hypotheses	Improvement at Posttest	Predicts	Improvement at Follow-Up
Hypothesis 1	Mindfulness	* 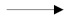 *	Mindful parenting
Hypothesis 2	Mindfulness	* 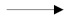 *	(Mental) health
Hypothesis 3	Mindful parenting	* 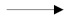 *	Parenting
Hypothesis 4	(Mental) health	* 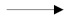 *	Parenting

**Table 2 ijerph-19-07571-t002:** Sociodemographic characteristic of the study participants.

	*Frequency*	*Percentage*
*Mothers*		
Ethnicity		
Dutch	42	68.9
European	3	4.9
Non-European	5	8.2
Mixed ethnicity	10	16.4
Unknown	1	1.6
Educational level		
Bachelor or master degree	38	62.3
Intermediate vocational education	11	18.0
High school	4	6.6
Unknown	8	3.2
Employment		
Full-time job	6	9.8
Part-time job	13	21.3
Stay-at-home mother	11	18.0
On sick leave	16	26.2
Maternity leave	6	9.8
Unknown	9	14.7
Relation to the child		
Biological parent	60	98.4
Foster parent	1	1.6
*Babies*		
Sex		
Boy	37	57.4
Girl	24	39.3
Unknown	2	3.3
Family situation		
With both parents	54	88.5
Primarily with mother	4	6.6
Primarily with father	1	1.6
Foster parents	1	1.6
Unknown	1	1.6

**Table 3 ijerph-19-07571-t003:** Means and standard deviations of all dependent measures at all measurement occasions, the Mindful with Your Baby training took place between pretest and posttest.

Outcome Variable	Waitlist		Pretest		Posttest		8-Week Follow-Up	
	*n*	M (SD)	*n*	M (SD)	*n*	M (SD)	*n*	M (SD)
*Mindfulness*								
Mindfulness	6	63.8 (13.6)	56	68.0 (11.7)	50	76.9 (12.4)	36	79.6 (13.3)
Mindful parenting	13	83.3 (14.3)	55	84.7 (12.0)	45	96.2 (12.4)	35	95.9 (13.0)
*(Mental) health*								
Psychological distress	13	84.3 (20.1)	54	72.5 (22.4)	43	64.1 (20.6)	27	59.4 (20.3)
Depressive mood	12	16.5 (3.7)	58	15.6 (3.6)	50	14.0 (3.9)	36	13.4 (3.5)
Physical health complaints	12	18.0 (4.6)	58	16.7 (4.0)	50	15.9 (4.8)	36	14.6 (3.8)
*Parenting*								
Parenting stress	12	45.4 (11.2)	58	44.2 (8.7)	50	39.5 (8.7)	36	38.7 (7.7)
Parental self-efficacy	11	80.0 (20.4)	53	92.6 (19.2)	32	101.4 (18.0)	29	102.6 (18.2)
Mother-infant bonding	13	10.8 (2.9)	57	11.6 (2.8)	50	12.6 (2.9)	36	13.3 (2.2)

Data are presented as mean (standard deviation).

**Table 4 ijerph-19-07571-t004:** Parameter estimates (standard error between brackets) followed by *t* values of multilevel.

		Intercept		Waitlist		Post-Test		8-Week Follow-Up	
	*n*	*β* (*SE*)	*t*	*β* (*SE*)	*t*	*β* (*SE*)	*t*	*β* (*SE*)	*t*
*Mindfulness*									
Mindfulness	57	−0.34 (0.12)	−3.73 **	−0.04 (0.04)	−0.96	0.68 (0.12)	5.61 **	0.85 (0.16)	5.40 **
Mindful parenting	56	−0.45 (0.12)	−3.86 **	−0.19 (0.10)	−1.88	0.81 (0.11)	7.48 **	0.79 (0.15)	5.15 **
*(Mental) health*									
Psychological distress	55	0.20 (0.14)	1.48	0.14 (0.09)	1.56	−0.34 (0.11)	−3.14 **	−0.56 (0.11)	−5.10 **
Depressive mood	59	0.26 (0.12)	2.14 *	0.18 (0.19)	0.92	−0.43 (0.13)	−3.19 **	−0.61 (0.17)	−3.48 **
Physical health complaints	59	0.18 (0.12)	1.43	0.06 (0.22)	0.28	−0.21 (0.16)	−1.34	−0.57 (0.13)	−4.25 **
*Parenting*									
Parenting stress	59	0.32 (0.13)	2.53	−0.07 (0.13)	−0.59	−0.54 (0.10)	−5.59 **	−0.66 (0.16)	−4.15 **
Parental self-efficacy	54	−0.18 (0.13)	−1.37	0.13 (0.12)	1.06	0.38 (0.11)	3.67 **	0.66 (0.17)	−3.90 **
Mother-infant bonding	58	−0.25 (0.13)	−1.91	−0.01 (0.13)	−0.04	0.41 (0.11)	3.72 **	0.66 (0.14)	4.65 **

Models of treatment outcome predicted by measurement occasion (deviations from pretest).*: *p* < 0.05, **: *p* < 0.01. *β* = Parameter estimate; can be interpreted as Cohen’s *d* effect size of change.

## Data Availability

The data presented in this study are available on request from the corresponding author. The data are not publicly available due to ethical reasons (personal data).

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
