# Peer review of "Do Improvements in Maternal Mental Health Predict Improvements in Parenting? Mechanisms of the Mindful with Your Baby Training"

_ijerph, 2022, doi:10.3390/ijerph19137571_

Round 1
Reviewer 1 Report
Please see the attachment

Author Response
Response to Reviewer 1
I appreciated the opportunity to read of your important work. The concept of mindful parenting is just emerging in large swaths of the US and your paper is an important contribution to supporting maternal and infant mental health. Overall, I found the manuscript to be easy to follow but made a few suggestions. I appreciated the attempts to discern directionality and interactions.
- Thank you for the time and effort spent in reviewing our manuscript. Thank you also for your valuable comments and for your kind words about our work.
Introduction
Should Mindful with your baby actually read Mindful with Your Baby, since it is the title of the program? (I recognize you have published under the first title already but it struck me as a title so I thought I would mention it).
- Thank you for this suggestion. We agree with you, and made this change throughout the manuscript.
What you consider to be mental illness should be defined earlier (i.e., depression, anxiety, and PTSD).
- We added the following definition to the first sentence of the second paragraph of the introduction: “Maternal mental health problems, defined as the presence of symptoms of stress, depression and anxiety, resulting in emotional suffering [8], can have negative consequences for child development, even if symptoms do not meet criteria for a disorder [8,9].”
Lines 64 -68 seem better placed in the next paragraph on parenting stress.
- Yes, we agree with your suggestion. We have moved this sentence according to your advice.
Line 97 – the concept of resilience has many definitions. It would help the reader to know what you mean by resilience.
- We added the following subsentence to the sentence in which resilience was mentioned: “which was defined in this study as tenacity, ability to tolerate negative affect, security of relationships, control, and spiritual influences)”
Line 133 – there is an “and” missing after overreactivity.
- Thank you for reading our manuscript so carefully. We have added ‘and’ to this sentence.
Page 3 – definitions of or distinctions between general mindfulness and mindful parenting would be helpful to the reader.
- We indeed added the following text to the paragraph in which mindful parenting was introduced: “Mindful parenting interventions extend mindfulness training, which focus merely on intrapersonal experiences, to the interpersonal context of parenting [46,47]. Mindfulness can be defined as “the awareness that emerges through paying attention, on purpose, in the present moment, and nonjudgmentally to the unfolding of experience moment by moment” [48], and mindful parenting as “paying attention to your child and your parenting in a particular way: intentionally, here and now, and non-judgmentally” [49].”
Materials and Methods
Page 5 – what defines highly educated?
- With highly educated we meant a bachelor or master degree. Because the sociodemographic characteristics have been moved to a table, we adjusted this in Table 1.
Line 243 needs editing Page 5 – the term “housewives” is outdated (at least in the US and upholds the concept of patriarchy. I would suggest a more descriptive term - i.e., did not work outside the home; stay-at-home mother).
- Thank you for pointing this out for us. We also adjusted this in Table 1.
What is the source of ‘the seven attitudes’ and ‘supporting your baby’s autonomy’?
- We added the sources for the readings to the text and the reference list.
Section 2.3
Line 269 – what do you mean by “certain practices?”
- We have explained this better by adding information to this particular sentence: “The other sessions do have a program with several exercises, but at the same time flexibility is built in, so that if stress arises in (one of) the mother-baby dyads, a short mindfulness meditation can be practiced with the group, namely a parent-child breathing space. This is an adjustment of the 3-minute breathing space [60], in which parents not only practice with allowing their own feelings, but also with allowing feelings of their baby.”
When you say “on location” – is that a clinic? A psychiatric facility? A community center?
- We clarified this issue by expanding the sentence as follows: “Before the COVID-19 pandemic, all trainings were given on location, which was an outpatient mental health care centre for the majority of the participants, or, for the mother-baby dyads who followed the training in a preventive setting, a mindfulness training centre, or a space rented by a mindfulness teacher.”
Section 2.4 – Measures
It might help an audience unfamiliar with mindfulness to have a sample question or two each for the mindfulness and the mindful parenting measures.
- We added the following examples of items of the IM-P and the FFMQ, respectively: “An example of an item was “I am often so busy thinking about other things that I realize I am not really paying attention to my baby” and “I tell myself I should not be feeling the way I am feeling.”
Why were the three parenting subscales combined for the OBVL?
- In order to limit the number of outcome measures, in general we decided to use the total scales instead of the subscales of the measures that we included (FFMQ, IM-P, OQ). The only exception was made for the OBVL, because in our opinion two of its’ subscales were more logical to categorize as a parental (mental) health outcome (depressive mood and physical health complaints) than as a parenting outcome. Therefore, we took these two subscales ‘apart’ from the rest of the scale, and computed a new ‘total scale’ on the basis of the other three subscales representing parenting stress-related outcomes. The alphas of this combined parenting stress scale were good to excellent (between .88 and .93).
On line 470 and 471 you say, “Improvement in mindful parenting between pretest and posttest was not found to be a significant predictor of parental stress (p = .057) or maternal bonding (p = .249) at follow-up” but on line 532 and 533 you say, “Improvement in mindful parenting between pretest and posttest predicted aspects of parenting, namely parenting stress and parenting self-efficacy, but not mother-to-infant bonding.” While it appears that the distinction is at follow up, this is confusing to the reader.
- We are very glad that you were able to detect this mistake. In earlier analyses, we did find a significant association between mindful parenting and parenting stress, but we forgot to adjust this part of the summery of the results in the discussion when it turned out that this association was no longer significant. We have now been able to correct our mistake, thank you.
Line 640 appears to be missing the word “be” in the sentence.
- Thank you again for your careful reading. The word ‘be’ has been added.
Given the complexity of the model and the interactions you are trying to capture, I found the overall discussion to be rich, reasonable, and theoretically well-grounded. It may be beyond the scope of your discussion of future directions but substantively, attending to observations of parental withdrawal (i.e., not just anger) is also of critical importance to the well-being of the baby. See work of Karlen Lyons-Ruth re: maternal withdrawal.
- Indeed, the work of Karlen Lyons-Ruth is of interest. We have added the following sentences to the future directions: “In a review on the pathways that are associated with maternal withdrawal, it was concluded that maternal withdrawal was a risk factor for later psychopathology in the children [89]. Earlier research on the effects of Mindful with Your Baby training on parent-child interaction did not include a measure of withdrawal [89].”
Line 723 should read professionals.
- We have indeed added an ‘s’ to ‘professional’, thank you.
Reviewer 2 Report
I congratulate the authors of this study for their work.
I consider the topic addressed in this study to be of great relevance: knowing and improving the mental health of mothers and its consequence in the parenting process, and therefore, the benefits that can be derived for the children/babies.
- Among the strengths, I want to highlight that this study replicates procedures and results of the improvement in maternal mental health and improvement of the parenting process.
On the other hand, it has tried to go beyond verifying the efficacy of a type of treatment, but rather tries to explain the mechanisms of change.
For all this, congratulations.
Among the suggestions for improving the text, I propose the following:
- In the Participants section, it is indicated that the mothers' way of accessing Mindful with your baby-training was following two procedures: Mainly through referral from a doctor or a psychologist (82%) and 18% through preventive setting. It is not clear what this means of access to the training program. Please clarify.
- In relation to this way of accessing the training program, were differences in the different psychological variables such as psychological distress and depressed mood analyzed?
- The study indicates the existence of a group on the waiting list. How was it decided who was on the waiting list? Did these mothers finally receive training as well?
- Perhaps it would be interesting to reduce the length of the article, to use a table to collect the sociodemographic characteristics of the participants (which appear in the paragraph between lines 227 and 247).
- I also think it would be very illustrative to include a Flowchart to collect information on the participants in the three evaluations.
- The Mindful with your baby program is made up of 8 sessions plus a follow-up session. What content is worked on in each session?
In the first session the topic 'Mindfully observing your baby'? And in the second 'Creating space for yourself'? In the third session ‘Closeness and distance’, and….. ‘Dealing with expectations’,… Is that right? What is worked on in the rest of the sessions? And in the follow-up session, 8 weeks later?
How long is each session?
- Finally, given that some participants received the training face-to-face and others virtually, has it been analyzed whether there are differences depending on the format of the sessions (virtual versus face-to-face) in the dependent variables?
Author Response
Response to Reviewer 2
I congratulate the authors of this study for their work.
I consider the topic addressed in this study to be of great relevance: knowing and improving the mental health of mothers and its consequence in the parenting process, and therefore, the benefits that can be derived for the children/babies.
- Among the strengths, I want to highlight that this study replicates procedures and results of the improvement in maternal mental health and improvement of the parenting process.
On the other hand, it has tried to go beyond verifying the efficacy of a type of treatment, but rather tries to explain the mechanisms of change.
For all this, congratulations.
- Thank you for your positive feedback about our manuscript. Thank you also for your valuable comments that has helped us improve the manuscript.
Among the suggestions for improving the text, I propose the following:
- In the Participants section, it is indicated that the mothers' way of accessing Mindful with your baby-training was following two procedures: Mainly through referral from a doctor or a psychologist (82%) and 18% through preventive setting. It is not clear what this means of access to the training program. Please clarify.
- Thank you for your question. I hope we interpreted your question in the right way, by adding the following information to the Participants subsection: “In case mothers were referred for the training, the costs were covered by the municipality or insurance. When mothers followed the training in a preventive setting, depending on the family income, they received a discount on, or refund of the training costs if they participated in the research project, which was funded by the MindMore Foundation.”
- In relation to this way of accessing the training program, were differences in the different psychological variables such as psychological distress and depressed mood analyzed?
- This is indeed an interesting question, that we are planning to answer later on in this ongoing study. We hope to finalize data collection in about a year from now. In this last year of data collection, we especially hope to include more participants in the preventive setting, so we will be able to test baseline differences as well as differences in change between pretest and posttest between the groups. We have added the following information to the Procedure subsection: “The data for this study was derived from an ongoing study, which focuses on the (comparison of) Mindful with Your Baby in a clinical and preventive setting.”
- The study indicates the existence of a group on the waiting list. How was it decided who was on the waiting list? Did these mothers finally receive training as well?
- Yes, all mothers received the training. Mothers were not put on a waiting list, but mothers who were referred or admitted to the training at least one month prior to the start of the training (and who thus had to wait at least one month for the training), received an extra measurement, namely the waitlist measurement. They also completed pretest measurement right before the start of the training. We clarified this issue by changing the text on the waitlist measurement as follows: “All measurement occasions consisted of questionnaires that could be filled in online. Only mothers who were referred or admitted to the training at least one month prior to the start of the training (and who thus had to wait at least one month for the training), were invited to complete a waitlist assessment, to control for factors such as time, participation in research, hope for change. . Thirteen (21.3%) participants completed waitlist assessment (average waiting time between the completion of the waitlist assessment and the start of the training was 45.9 days, sd = 13.5, range 19-67 days).”
- Perhaps it would be interesting to reduce the length of the article, to use a table to collect the sociodemographic characteristics of the participants (which appear in the paragraph between lines 227 and 247).
- We added a table with the sociodemographic characteristics. Although the length of the article was not reduced by adding a table with the sociodemographic characteristics, we think that the table did bring more structure in the presentation of the sociodemographic characteristics.
- I also think it would be very illustrative to include a Flowchart to collect information on the participants in the three evaluations.
- Thank you for your suggestion. We added a flowchart to the manuscript.
- The Mindful with your baby program is made up of 8 sessions plus a follow-up session. What content is worked on in each session?
- We added the following text to the Intervention subsection: “Each sessions starts with a 10- to 15-minute formal meditation practice, such as a sitting meditation with attention for the breath and body, followed by inquiry. This inquiry is focused on experiences during the meditation, among which the way in which the parent divided her attention between herself and her baby. After the inquiry, experiences with the homework is discussed. In some sessions, there is another short practice before the break, such as the 3-minute breathing space meditation. In the break, there is space for the mothers to talk with each other, drink something, and for example to feed their baby. After the break, an exercise is done that is focused on the theme of the session, such as visualisation on a moment that the mother felt a distance in the contact between herself and her baby, and on how the mother can reconnect with her baby again after such a moment. The last exercise of the sessions with the babies present is a watching meditation, in which mothers practice to observe their baby with full attention and with curiosity, to also notice what they experience while doing this, to reflect on what the baby might be experiencing, and some meditation instructions are also given around the theme of the session.” (…) “A table which summarizes the main and secondary theme, the practices, home practices and readings of each session has been published in a previous study on Mindful with your baby [62].”
In the first session the topic 'Mindfully observing your baby'? And in the second 'Creating space for yourself'? In the third session ‘Closeness and distance’, and….. ‘Dealing with expectations’,… Is that right? What is worked on in the rest of the sessions? And in the follow-up session, 8 weeks later?
- Actually, these examples were the topics of session 2, session 3, session, 6, and session 7. We have now included the topics of all sessions: “Each session has its own theme: 1) Becoming aware of the automatic pilot, 2) Mindfully observing your baby, 3) Creating space for yourself, 4) Responding sensitively to your baby, 5) Taking care of yourself in difficult moments, 6) Closeness and distance, 7) Dealing with expectations, and 8) Mindful parenting: each time, beginning anew”. (…) “In the follow-up session, the mothers are reminded of all themes of the previous sessions.”
How long is each session?
- Each session is 2 hours. This information was added to line 265.
- Finally, given that some participants received the training face-to-face and others virtually, has it been analyzed whether there are differences depending on the format of the sessions (virtual versus face-to-face) in the dependent variables?
- We agree that this will be interesting to study. We aim to study differences between the face-to-face and online trainings in a paper that we will write after the data collection of this ongoing study has been finalized. We added the following sentence to the paragraph on future directions: “Future research should also address the differences in effect between the face-to-face and online version of Mindful with Your Baby.”
Reviewer 3 Report
I found the article to be well written, clearly presented and interesting. However, the text is long and complex because it involves comparisons between many variables and at different time points. This, coupled with the font size makes it difficult to read.
I have two suggestions to ease reading and comperhension.
1) To include a CONSORT Flow Diagram, of the progress through the phases of the trial (that is, enrolment, intervention allocation, follow-up, and data analysis).
2) To add a figure with a model of the hypothesised relationships of the variables over time.
On page 12, line 563, reference 76 does not correspond to the text, it should be 77. I did not check the other references, but I suggest checking all.
Author Response
Response to reviewer 3.
I found the article to be well written, clearly presented and interesting. However, the text is long and complex because it involves comparisons between many variables and at different time points. This, coupled with the font size makes it difficult to read.
I have two suggestions to ease reading and comprehension.
- Thank you for your compliments, and your constructive feedback on how to improve comprehension of our study. This has helped us to improve our manuscript.
- To include a CONSORT Flow Diagram, of the progress through the phases of the trial (that is, enrolment, intervention allocation, follow-up, and data analysis).
- Thank you for your suggestion. We included a CONSORT flow diagram.
- To add a figure with a model of the hypothesised relationships of the variables over time.
- We thought this was a very good idea, and added both a table with the different hypotheses and time points, and one figure that that integrates all hypothesized relationships.
On page 12, line 563, reference 76 does not correspond to the text, it should be 77. I did not check the other references, but I suggest checking all.
- You are right, there were some mistakes in the references. We have now checked them all.